# From behavior to circuit modeling of light-seeking navigation in zebrafish larvae

Sophia Karpenko[1,2], Sebastien Wolf[3,4], Julie Lafaye[1], Guillaume Le Goc[1], Thomas Panier[1], Volker Bormuth[1], Raphaël Candelier[1], Georges Debrégeas[1]*

[1]Sorbonne Université, CNRS, Institut de Biologie Paris-Seine (IBPS), Laboratoire Jean Perrin (LJP), Paris, France; [2]Université Paris Sciences et Lettres, Paris, France; [3]Laboratoire de Physique de l'Ecole Normale Supérieure, CNRS UMR 8023 & PSL Research, Paris, France; [4]Institut de Biologie de l'Ecole Normale Supérieure, CNRS, INSERM, UMR 8197 & PSL Research, Paris, France

**Abstract** Bridging brain-scale circuit dynamics and organism-scale behavior is a central challenge in neuroscience. It requires the concurrent development of minimal behavioral and neural circuit models that can quantitatively capture basic sensorimotor operations. Here, we focus on light-seeking navigation in zebrafish larvae. Using a virtual reality assay, we first characterize how motor and visual stimulation sequences govern the selection of discrete swim-bout events that subserve the fish navigation in the presence of a distant light source. These mechanisms are combined into a comprehensive Markov-chain model of navigation that quantitatively predicts the stationary distribution of the fish's body orientation under any given illumination profile. We then map this behavioral description onto a neuronal model of the ARTR, a small neural circuit involved in the orientation-selection of swim bouts. We demonstrate that this visually-biased decision-making circuit can capture the statistics of both spontaneous and contrast-driven navigation.

*For correspondence: georges.debregeas@upmc.fr

Competing interests: The authors declare that no competing interests exist.

## Introduction

Animal behaviors are both stereotyped and variable: they are constrained at short time scale to a finite motor repertoire while the long-term sequence of successive motor actions displays apparent stochasticity. This dual characteristic is immediately visible in the locomotion of small animals such as Nematodes (*Stephens et al., 2008*), Zebrafish (*Girdhar et al., 2015*) or *Drosophila* larvae (*Gomez-Marin and Louis, 2012*), which consists of just a few stereotyped maneuvers executed in a sequential way. In this case, behavior is best described as a set of statistical rules that defines how these elemental motor actions are chained. In the presence of sensory cues, two types of behavioral responses can be distinguished. If they signal an immediate threat or reward (e.g. the presence of a predator or a prey), they may elicit a discrete behavioral switch as the animal engages in a specialized motor program (e.g. escape or hunt, *Budick and O'Malley, 2000*; *Fiser et al., 2004*; *Bianco et al., 2011*; *McClenahan et al., 2012*; *Bianco and Engert, 2015*). However, most of the time, sensory cues merely reflect changes in external factors as the animal navigates through a complex environment. These weak motor-related cues interfere with the innate motor program to cumulatively promote the exploration of regions that are more favorable for the animal (*Tsodyks et al., 1999*; *Fiser et al., 2004*).

A quantification of sensory-biased locomotion thus requires to first categorize the possible movements, and then to evaluate the statistical rules that relate the selection of these different actions to the sensory and motor history. Although the probabilistic nature of these rules generally precludes a

**eLife digest** All animals with the ability to move use sensory signals to help them navigate towards areas that seem better than their current location. Such areas might contain desirable things like food and mates, or they might allow an animal to escape from threats such as predators. But how the brain gives rise to this navigation behavior is unclear.

Karpenko et al. have now obtained insights into the underlying mechanism by studying a behavior in zebrafish larvae called phototaxis. Phototaxis is the tendency to move in response to light. The advantage of using zebrafish larvae to study this behavior is that their brains are small and semi-transparent. This makes it possible to record the activity of almost every neuron. As a result, an individual's brain activity can be mapped on to their behavior more precisely than in most other species.

To probe how visual cues influence fish behavior, Karpenko et al. exposed individual fish to a carefully controlled virtual light source and then tracked their movements with a camera. The fish used two strategies to move towards the light. They selected their next movement based partly on the difference in the amount of light reaching each of their eyes, and partly on the change in overall brightness with each swim movement. Karpenko et al. used this information to build a numerical model of fish phototaxis, and to show how a simple brain circuit could generate this behavior.

Species whose brains differ in size and structure may nevertheless develop similar strategies to perform similar tasks. By quantifying a generic behavior in a simple animal model, this study could provide insights into comparable behaviors in other species. In addition, the study suggests a simple mechanism for how animals select actions on the basis of sensory signals, which may also be relevant to other species and other tasks.

deterministic prediction of the animal's trajectory, they may still provide a quantification of the probability distribution of presence within a given environment after a given exploration time.

In physics terms, the animal can thus be described as a random walker, whose transition probabilities are a function of the sensory inputs. This statistical approach was originally introduced to analyze bacteria chemotaxis (*Lovely and Dahlquist, 1975*). Motile bacteria navigate by alternating straight swimming and turning phases, so-called runs and tumbles, resulting in trajectories akin to random walks (*Berg and Brown, 1972*). Chemotaxis originates from a chemical-driven modulation of the transition probability from run to tumble: the transition rate is governed by the time-history of chemical sensing. How this dependency is optimized to enhance gradient-climbing has been the subject of extensive literature (*Macnab and Koshland, 1972*; *Adler and Tso, 1974*; *Mello and Tu, 2007*; *Yuan et al., 2010*; *Celani and Vergassola, 2010*). More recently, similar descriptions have been successfully used to quantify chemotaxis and phototaxis in multicellular organisms such as *Caenorhabditis elegans* (*Ward, 1973*; *Miller et al., 2005*; *Ward et al., 2008*), Drosophila larva (*Sawin et al., 1994*; *Kane et al., 2013*; *Gomez-Marin et al., 2011*; *Tastekin et al., 2018*) or different types of slugs (*Matsuo et al., 2014*; *Marée et al., 1999*). Although the sensorimotor apparatus of these animals are very different, the taxis strategies at play appear to be convergent and can be classified based on the gradient-sensing methods (*Fraenkel and Gunn, 1961*; *Gomez-Marin and Louis, 2012*). Tropotaxis refers to strategies in which the organism directly and instantaneously infers the stimulus direction by comparison between two spatially distinct sensory receptors. In contrast, during klinotaxis, the sensory gradient is inferred from successive samplings at different spatial positions. This second strategy is particularly adapted when the organism has only one receptor, or if the sensory gradient across the animal's body is too small to be detected (*Humberg et al., 2018*). It requires at least a basic form of memory, since the sensory information needs to be retained for some finite period of time.

In the present work, we implement such a framework to produce a comprehensive statistical model of phototaxis in zebrafish larvae. Zebrafish larva is currently the only vertebrate system that allows in vivo whole-brain functional imaging at cellular resolution (*Panier et al., 2013*; *Ahrens et al., 2013*). It thus provides a unique opportunity to study how sensorimotor tasks, such as sensory-driven locomotion, are implemented at the brain-scale level.

Although adult zebrafish are generally photophobic (or scototactic, *Serra et al., 1999*; *Maximino et al., 2007*), they display positive phototaxis at the larval stage, from 5 days post-fertilization (dpf) on (*Orger and Baier, 2005*). At this early stage, their locomotion consists of a series of discrete swimming events interspersed by ~1 s long periods of inactivity (*Girdhar et al., 2015*). Previous studies have shown that, when exposed to a distant light source, the first bouts executed by the fish tend to be orientated in the direction of the source (tropotaxis) (*Burgess et al., 2010*). Furthermore, *Chen and Engert (2014)* have shown, using a virtual reality assay, that zebrafish are able to confine their navigation within a bright region in an otherwise dark environment even when deprived from stereovisual contrast information. This latter study thus established that their phototactic behavior also involves a spatio-temporal integration mechanism (klinotaxis).

From a neuronal viewpoint, recent calcium imaging experiments identified a small circuit in the rostral hindbrain that plays a key role in phototaxis (*Ahrens et al., 2013*; *Dunn et al., 2016*; *Wolf et al., 2017*). This region, called ARTR (anterior rhombencephalic turning region) or HBO (hindbrain oscillator), displays pseudo-periodic antiphasic oscillations, such that the activity of the left and right subpopulations alternate with a ~20 s period. This alternation was shown to set the coordinated direction of the gaze and tail bout orientation, thus effectively organizing the temporal sequence of the successive reorientations. It was further shown that this circuit oscillation could be driven by whole-field illumination of the ipsilateral eye, such as to favor the animal's orientation towards a light source (*Wolf et al., 2017*).

In the present study, we aim at quantifying the statistical rules that control the larva's reorientation dynamics in the presence of a continuous angular gradient of illumination (orientational phototaxis). Using a virtual-reality closed-loop assay, we quantify how swim bouts selection is statistically controlled by the light intensity received on both eyes prior to the bout initiation, or the change in illumination elicited by the previous swim bout. Our experimental configuration allows us to disentangle the contribution of the two aforementioned strategies: tropotaxis and klinotaxis. From the analysis of this short-term behavior, we built a minimal Markov model of phototaxis, from which we compute the long-term distribution of orientations for any angular profile of illumination. This model offers explicit predictions of the statistics of the fish orientation that quantitatively compare with the experimental observations. We further expand on a recent rate model of the ARTR circuit to propose a functional neuronal model of spontaneous navigation and contrast-biased orientation selection. We demonstrate that the statistics of turn orientation can be fully understood by assuming that this self-oscillating circuitry, that selects the orientation of turning bouts, integrates stereovisual contrast in the form of incoming currents proportional to the visual stimulus.

## Results

### Kinematics of spontaneous navigation as a first-order autoregressive process

Zebrafish larvae aged 5–7 dpf were placed one at a time in a Petri dish (14 cm in diameter). Their center-mass position and body axis orientation were tracked in real time at 35 frames/s (*Figure 1A–B*). This information was used to deliver a body-centered visual stimulus using a video-projector directed onto a screen supporting the Petri dish.

Prior to each phototactic assay, the larva was allowed an $\approx 8$ min-long period of spontaneous exploration under uniform and constant illumination at maximum intensity $I_{max} = 450\mu W.cm^{-2}$. Such pre-conditioning phases were used to promote light-seeking behavior (*Burgess and Granato, 2007*), while enabling the quantification of the basal exploratory kinematics for each fish.

Larval zebrafish navigation is comprised of discrete swim bouts lasting $\approx 100ms$ and interspersed with 1 to 2s-long inter-bout intervals ($\tau_n$) during which the fish remains still (*Dunn et al., 2016*). Each bout results in a translational motion of the animal and/or a change in its body axis orientation, and can thus be automatically detected from kinematic parameters. As we are mostly interested in the orientational dynamics, we extracted a discrete sequence of orientations $\alpha_n$ measured just before each swimming event $n$ (*Figure 1B–C*) from which we computed the bout-induced reorientation angles $\delta\alpha_n = \alpha_{n+1} - \alpha_n$.

Although the complete swim bouts repertoire of zebrafish larvae is rich and complex (*Johnson et al., 2019*), the statistical distribution of the reorientation angles $P(\delta\alpha_n)$ in such unbiased

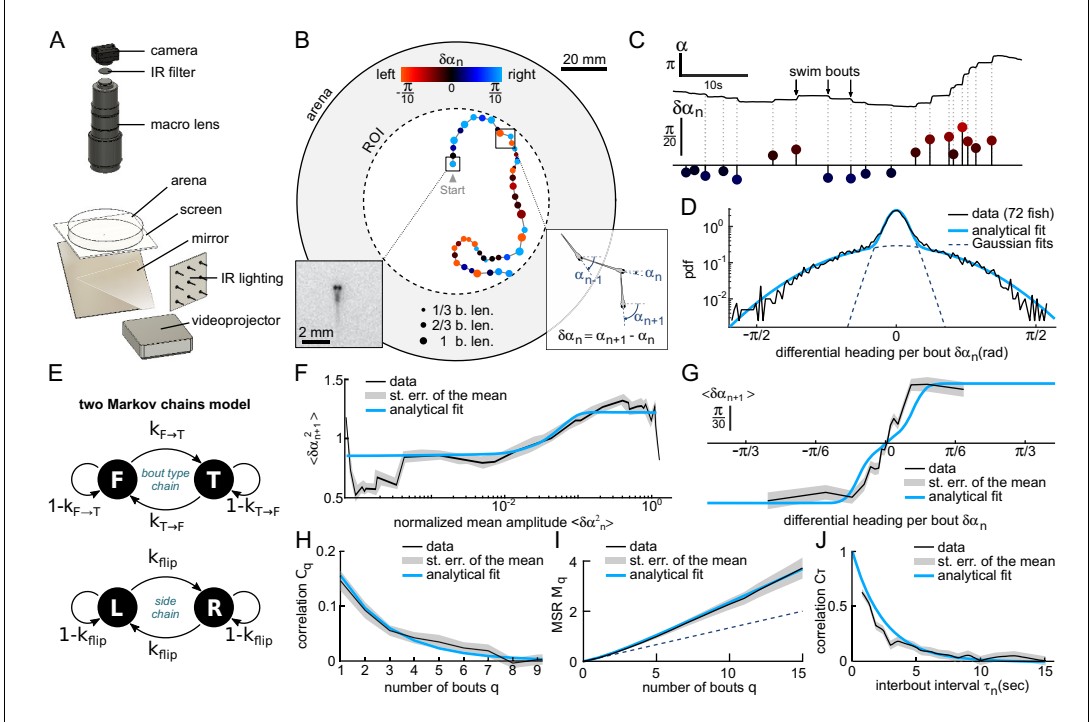

**Figure 1.** Kinematics of spontaneous navigation. $N = 75$ fish, $n = 16,147$ bouts, mean of 7 trajectories per fish. (A) Experimental setup: real-time monitoring of the larva's position and orientation using IR illumination, enables closed-loop visual stimulation using a video projector. (B) Typical trajectory of a 6 days old larva in the region of interest (ROI) of the arena under constant, uniform illumination. Each point indicates the fish position at the onset of a swim bout. Dots' size and color encode the bout distance and bout reorientation angle, respectively. Insets: blow-up of an example frame (left) and definition of the reorientation angle $\delta\alpha_n$ at bout index $n$ (right). b.len: body length. (C) Time-sequence of the fish body orientation $\alpha$ (top). Swim bouts elicit rapid re-orientations. The angular dynamics can thus be represented as a series of discrete reorientation events of various amplitudes $\delta\alpha_n$ (color code as in (B)). (D) Experimental (dark) and analytical (blue) distributions (pdf: probability density function) of reorientations $\delta\alpha_n$. The two normal distributions used in the fit with *Equation A1*, weighted by $p_{turn}$ and $1 - p_{turn}$, are also displayed in dashed blue lines. (E) Two independent Markov chains model for spontaneous navigation: the *bout type* chain controls the forward scoot (F) versus turning (T) state, with transitions rates $k_{T\to F}$ and $k_{F\to T}$. The *side* chain controls the transitions between left (L) and right (R) headings when the animal is in the turning state, with transition rate $k_{flip}$. (F) Mean squared reorientation amplitude of bout $n + 1$ as a function of the squared amplitude of bout $n$ (grey), and its associated analytical fit (blue, Appendix From behavior to circuit modeling of light-seeking navigation in zebrafish larvae *Equation A5*). (G) Average reorientation of bout $n + 1$ as a function of the reorientation at bout $n$ (grey), and its associated analytical fit (blue, *Equation A11*). (H) Correlation in reorientation angles $C_q$ as a function of the number of bouts (grey) and associated fit (blue, *Equation A14*). (I) Mean square reorientation (MSR) $M_q$ as a function of the number of bouts, and associated fit (blue, *Equation A17*). The dotted line is the linear extrapolation of the first two data points and corresponds to the diffusive process expected for a memory-less random walk (no correlation in bout orientation). (J) Orientation correlation of turning bouts (thresholded at 0.22rad) as a function of the time elapsed between those bouts. The blue line is the exponential fit. Data from this and the following figures are available at *Karpenko (2019a)* (copy archived at https://github.com/elifesciences-publications/programs_closed-loop_phototaxis).

conditions can be correctly captured by the weighted sum of two zero-mean normal distributions, $P(\delta\alpha_n) = p_{turn}\mathcal{N}(0, \sigma^2_{turn}) + p_{fwd}\mathcal{N}(0, \sigma^2_{fwd})$, reflecting the predominance of only two distinct bouts types: turning bouts (standard deviation $\sigma_{turn} = 0.6$) and forward scoots ($\sigma_{fwd} = 0.1$) (*Figure 1D*). This bimodal distribution is consistent with the locomotor repertoire of larvae described by *Marques et al. (2018)* during spontaneous swimming and phototactic tasks. In the absence of a visual bias, the turning bouts and forward scoots were found to be nearly equiprobable, $p_{turn} = 1 - p_{fwd} = 0.41$.

Successive bouts were found to exhibit a slightly positive correlation in amplitude (*Figure 1F*). This process can be captured by a two-state Markov-chain model that controls the alternation between forward and turning bouts, while the amplitude within each population is randomly sampled from the corresponding distribution (*Figure 1E*). Within this scheme, we analytically derived the dependence in the amplitude of successive bouts and thus estimated the forward-to-turn and turn-to-forward transition rates, noted $k_{f\to t}$ and $k_{t\to f}$ (all analytical derivations are detailed in Appendix

From behavior to circuit modeling of light-seeking navigation in zebrafish larvae). We found that $k_{f \to t}/p_{turn} = k_{t \to f}/p_{fwd} \approx 0.8$. This indicates that the probability to trigger a turn (*resp.* forward) bout is decreased by only 20% if the previous bout is a forward (*resp.* turn) bout. For the sake of simplicity, we ignore in the following this modest bias in bout selection and assume that the chaining of forward and turning bout is memory-less by setting $k_{f \to t} = p_{turn}$ and $k_{t \to f} = p_{fwd}$. We checked, using numerical simulations, that this simplifying assumption has no significant impact on the long-term navigational dynamics: the results presented in the following, notably the diffusion coefficient, remain essentially unchanged when this small correlation in bout type selection is taken into account.

In line with previous observations (*Chen and Engert, 2014*; *Dunn et al., 2016*), we also noticed that successive turning bouts tended to be oriented in the same (left or right) direction (*Figure 1G*). This orientational motor persistence was accounted for by a second Markov chain that set the orientation of turning bouts, and was controlled by the rate of flipping direction noted $k_{flip}$ (*Figure 1E* bottom). Notice that, in contrast with the model proposed by *Dunn et al. (2016)*, although the orientational state is updated at each bout, it only governs the direction of turning bouts. When a forward bout is triggered, its orientation is thus unbiased.

This model provides an analytical prediction for the mean reorientation angle $\langle \delta \alpha_n \rangle_{|\delta \alpha_{n-1}}$ at bout $n$ following a reorientation angle $\delta \alpha_{n-1}$ at bout $n-1$. This expression was used to fit the experimental data (*Figure 1G*) and allowed us to estimate the flipping rate $p_{flip} = 0.19$ (99% confidence bounds ±0.017). We further computed the autocorrelation function of the reorientation angles and the Mean Square Reorientation (*MSR*) accumulated after $n$ bouts (*Figure 1H–I*). Both were consistent with their experimental counterparts. In particular, this model quantitatively captures the ballistic-to-diffusive transition that stems from the directional persistence of successive bouts (*Figure 1I*). As a consequence, the effective rotational diffusivity at long time $D_{eff} = 0.3\, rad^2$ is about twice as large than the value expected for a memory-less random walk (i.e. with $p_{flip} = 0.5$, see dashed line in *Figure 1I*).

In this discrete Markov-chain model, time is not measured in seconds but corresponds to the number of swim bouts. It thus implicitly ignores any dependence of the transition rates with the interbout interval. We examined this hypothesis by evaluating the correlation in bouts orientations as a function of the time elapsed between them. To do so, we first sorted the turning bouts by selecting the large amplitude events ($|\delta \alpha| < 0.22\, rad$). We then binarized their values, based on their leftward or rightward orientation, yielding a discrete binary signal $s(t_n) = \pm 1$. We finally computed the mean product $\langle s(t_n)s(t_p) \rangle$ for various time intervals $\Delta t = t_p - t_n$. The resulting graph, shown in *Figure 1J*, demonstrates that the correlation in orientation of successive bouts decays quasi-exponentially with the inter-bout period. This mechanism can be captured by assuming that the orientation selection at each bout is governed by a hidden two-state continuous-time process. The simplest one compatible with our observations is the telegraph process, whose transition probability over a small interval $dt$ reads $k_{flip}dt$, and whose autocorrelation decays as $\exp(-2k_{flip}t)$. Setting $k_{flip} = p_{flip}/median(\tau_n) = 0.2\, s^{-1}$, this model correctly captures the $\tau_n$-dependence of the orientational correlation of bouts.

In the two following sections, we use the discrete version of the Markov-chain model to represent the fish navigation, and investigate how the model parameters are modulated in the presence of a virtual distant light source. We then go back to the underlying continuous-time process when introducing a neuronal rate model for the orientation selection process.

## Contrast-driven phototaxis can be described as a biased random walk

We first examined the situation in which the perceived stereo-visual contrast is the only cue accessible to the animal to infer the direction of the light source (tropotaxis regime). The visual stimulus consisted of two uniformly lit half-disks, each covering one visual hemifield. The intensity delivered to the left and right eyes, noted $I_L$ and $I_R$ respectively, were locked onto the fish's orientation $\theta$ relative to the virtual light source (*Figure 2A*): the total intensity ($I_L + I_R$) was maintained constant while the contrast $c = I_L - I_R$ was varied linearly with $\theta$, with a mirror symmetry at $\pi/2$ (*Figure 2B*). This dependence was chosen to mimic the presence of a distant source located at $\theta = 0$ for which the contrast is null.

The orientation of the virtual source in the laboratory frame of reference was randomly selected at initiation of each assay. After only a few bouts, the animal orientation was found to be statistically

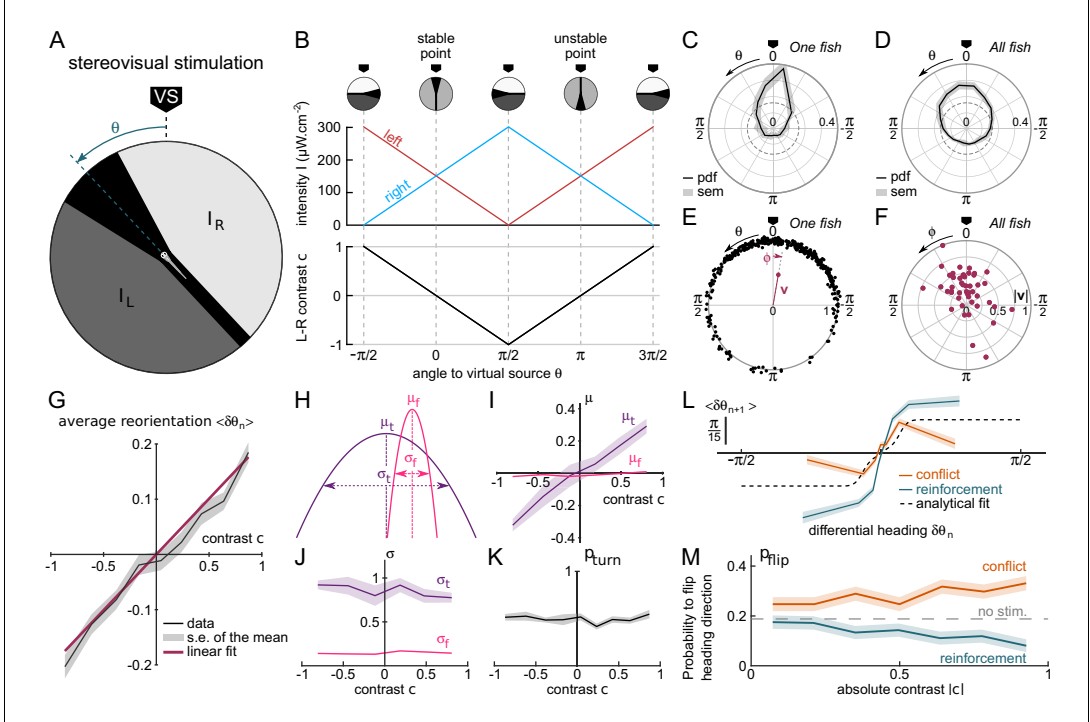

**Figure 2.** Contrast-driven phototaxis as a biased random walk. $N = 47$ fish, 18, 322 bouts, mean of 13 trajectories per fish. All statistical analyses are performed on the first 17 bouts, the first one excluded, for each assay. VS : virtual source. (**A**) Stimulus pattern delivered to the larva. The orientation relative to the virtual source is noted $\theta$. (**B**) Left and right intensities (top panel) and contrast $c = \frac{I_L - I_R}{I_L + I_R}$ (bottom panel) as a function of $\theta$. The virtual light source is defined by a null contrast ($c = 0$) and corresponds to a stable point ($\frac{dc}{d\theta} < 0$). (**C**) Probability density function (pdf) of orientations relative to the virtual light source for one fish during 20 trials, bouts 2 to 17 (n = 320 bouts). (**D**) Probability density function (pdf) of orientations for all tested fish ($N = 47$). Significantly biased toward the virtual source (V-test for non-uniformity with specified mean 0, $p_{val} < 10^{-11}$) (**E**) Definition of the mean resultant vector v for one fish. The points represent the angular positions $\theta_n$ of the fish relative to the source. The vector v is defined as $\mathbf{v} = \left| \frac{1}{N} \sum \exp i\theta_n \right|$. The mean angle to the source is $\phi = \arg(\mathbf{v})$ (**F**) Resultant vectors v for individual fish. (**G**) Mean reorientation $<\delta\theta_n>$ per bout as a function of contrast $c$ for all fish. Error bars represent the standard error of the mean. Red line is the linear fit with slope 0.2 rad. (**H**) Illustration of the shift in turning distribution ($\mu_t < 0$) induced by a negative contrast. (**I**) Means, (**J**) standard deviations and (**K**) relative weight of the turning distribution as a function of the contrast. For each value of the contrast, these quantities were extracted by double-Gaussian fitting of the bout angles. The error bars represent the 99% confidence interval from the fit. (**L**) Average reorientation at bout $n + 1$ as a function of the reorientation at bout $n$ in reinforcing (contrast and previous bout orientation are consistent) or conflicting (contrast and previous bout orientation are in conflict) situations. The dashed line is the analytical prediction in the absence of stimulation. (**M**) Probability of switching direction $p_{flip}$ as a function of the contrast, in situations of conflict or reinforcement. The online version of this article includes the following figure supplement(s) for figure 2:

**Figure supplement 1.** Evolution of the mean resultant vector projected on the direction of the virtual light source with the bout index.

**Figure supplement 2.** Evolution of contrast-driven bias slope with the bout index.

biased towards $\theta = 0$, as shown in *Figure 2C–D*. This bias was quantified by computing the population resultant v defined as the vectorial mean of all orientations (*Figure 2E*).

Trajectories that are strongly biased towards the source tend to exit the ROI earlier than unbiased trajectories, which are more tortuous and thus more spatially confined. This generates a progressive selection bias as the number of bouts considered is increased, as revealed by the slow decay of the resultant vector length (*Figure 2—figure supplement 1*). In order to mitigate this selection bias, all analyses of stationary distributions were restricted to bout indices lower than the median number of bouts per trial ($N \leq 17$), and excluding the first bout. Under this condition, we found that ~77% of zebrafish larvae display a significant phototactic behavior (*Figure 2D–F*, test of significance based on a combination of two circular statistic tests, *see* Materials and methods), a fraction consistent with values reported by *Burgess et al. (2010)* in actual phototactic assays .

From these recordings, we could characterize how the contrast experienced during the inter-bout interval impacts the statistics of the forthcoming bout. *Figure 2G* displays the mean reorientation

$\langle\delta\theta\rangle$ as a function of the instantaneous contrast $c$. This graph reveals a quasi-linear dependence of the mean reorientation with $c$, directed toward the brighter side. Notice that the associated slope shows a significant decrease in the few first bouts, before reaching a quasi-constant value (*Figure 2—figure supplement 2*). This effect likely reflects a short term habituation mechanism as the overall intensity drops by a factor of 2 at the initiation of the assay.

For a more thorough analysis of the bout selection mechanisms leading to the orientational bias, we examined, for all values of the contrast, the mean and variance of the two distributions associated with turning bouts and forward scoots, as well as the fraction of turning bouts $p_{turn}$ (*Figure 2H–K*). We found that the orientational drift solely results from a probabilistic bias in the selection of the turning bouts (left *vs* right) orientation: the mean orientation of the turning bouts varies linearly with the imposed contrast (*Figure 2I*). Reversely, the ratio of turning bouts and the variance of the two distributions are insensitive to the contrast *Figure 2J–K*). These results indicate that the stereo-visual contrast has no impact neither on bout type selection nor on bout amplitude.

As discussed in the preceding section, in the absence of visual cue, successive bouts tend to be oriented in the same direction. During phototaxis, the selection of the turning orientation is thus expected to reflect a competition between two distinct mechanisms: motor persistence, which favors the previous bout orientation, and stereo-visual bias, which favors the brighter side. To investigate how these two processes interfere, we sorted the bouts into two categories. In the first one, called 'reinforcement', the bright side is in the direction of the previous bout, such that both the motor and sensory cues act in concert. In the second one, called 'conflicting', the contrast tends to evoke a turning bout in a direction opposite to the previous one. For each category, we plotted the mean reorientation angle at bout $n$ as a function of the reorientation angle at bout $n-1$ (*Figure 2L*). We further estimated, for each condition and each value of the contrast, the probability of flipping orientation $p_{flip}$ (*Figure 2M* and Appendix 2). These two graphs show that the stereo-visual contrast continuously modulates the innate motor program by increasing or decreasing the probability of flipping bout orientation from left to right and vice versa. Noticeably, in the conflicting situation at maximum contrast, the visual cue and motor persistence almost cancel each other out such that the mean orientation is close to ($p_{flip} \sim 0.4$).

## Phototaxis under uniform stimulation is driven by a modulation of the orientational diffusivity

We now turn to the second paradigm, in which the stereo-visual contrast is null (both eyes receive the same illumination at any time), but the total perceived illumination is orientation-dependent (klinotaxis regime). We thus imposed a uniform illumination to the fish whose intensity $I$ was locked onto the fish orientation $\theta$ relative to a virtual light source. We tested three different illumination profiles $I(\theta)$ as shown in *Figure 3A*: a sinusoidal and two exponential profiles with different maxima. Despite the absence of any direct orientational cue, a large majority of the larvae (78%) displayed positive phototactic behavior: their orientational distribution showed a significant bias towards the virtual light source, that is the direction of maximum intensity (*Figure 3B–E*).

Although the efficiency of the phototactic behavior is comparable to the tropotaxis case previously examined, here we did not observe any systematic bias of the reorientation bouts towards the brighter direction (*Figure 3F*). This indicates that the larvae do not use the change in intensity at a given bout to infer the orientation of the source in order to bias the orientation of the forthcoming turn. Instead, the phototactic process originates from a visually driven modulation of the orientational diffusivity, as measured by the variance of the bout angle distributions (*Figure 3G*). The use of different profiles allowed us to identify which particular feature of the visual stimulus drives this modulation. Although the bout amplitude variance was dependent on the intensity $I$ and intensity change $\delta I$ experienced before the bout, these relationships were found to be inconsistent across the different imposed intensity profiles. In contrast, when plotted as a function of $\delta I/I$, all curves collapse (*Figure 3—figure supplement 1*). This observation is in line with the Weber-Fechner law (*Fechner, 1860*), which states that the perceived change scales with the relative change in the physical stimulus. One noticeable feature of this process is that the modulation of the turning amplitude is limited to illumination *decrement* (i.e. negative values of $\delta I/I$). In the terminology of bacterial chemotaxis (*Oliveira et al., 2016*), the zebrafish larva can thus be considered as a 'pessimistic' phototactic animal: the orientational diffusivity increases in response to a decrease in illumination

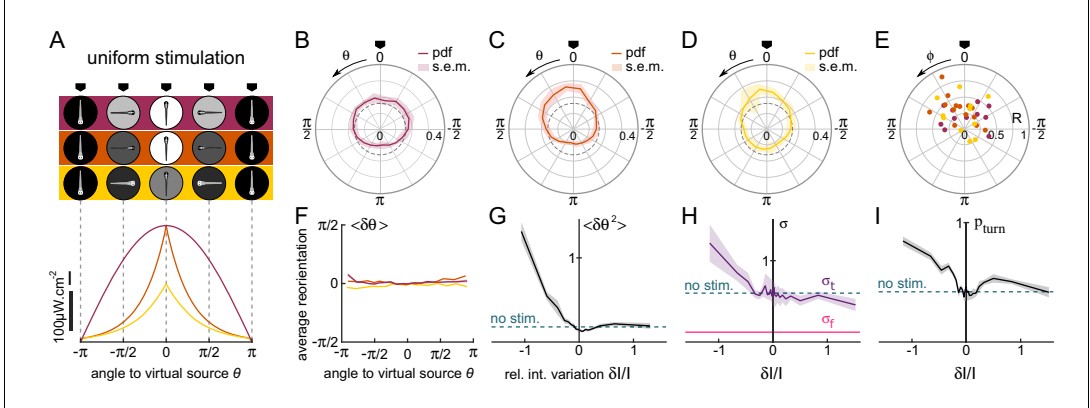

**Figure 3.** Orientational phototaxis driven by modulation of the global illumination. $N = 37$ fish, $n = 26,443$ bouts, mean of 23 trajectories per fish. **(A)** Top panel : Angle-dependent intensity profiles delivered to the larva. The virtual light source is located at $\theta = 0$, defined as the point of maximum intensity. The profiles are sinusoidal (uniform 1, purple) or exponentially shaped (uniform 2 and 3 orange and yellow, respectively). All statistics were computed using bout index two to the median number of bouts per sequence (resp. 27, 17 and 15 for the three profiles). **(B–D)** PDF of the fish orientations for the three profiles. All three distributions are significantly biased towards the virtual source (V-test for non-uniformity of circular data with specified mean , $p_{vals}$ respectively $9.10^{-3}$, $2.10^{-7}$ and $3.10^{-5}$).**(E)** Resultant vector v for all individual fish. **(F)** Mean reorientation per bout $<\delta\theta>$ of all fish as a function of $\theta$ for the three profiles. No significant bias towards the source ($\theta = 0$) is observed. **(G)** Variance of the reorientation angles $<\delta\theta^2>$ as a function of the relative change in intensity experienced at the previous bout $\delta I/I$. Error bars are standard error of the mean. **(H)** Standard deviation $\sigma_{turn}$ of turning bouts as a function of $\delta I/I$. The standard deviation of forward scouts was set at $\sigma_{fwd}$, and $\sigma_{turn}$ was then estimated using a double-Gaussian fitting of the bout angles. Error bars are the 99% confidence interval from fit. **(I)** Probability of triggering a turning bout as a function of $\delta I/I$. Error bars are the 99% confidence interval from the fit.

The online version of this article includes the following figure supplement(s) for figure 3:

**Figure supplement 1.** Variance of $\delta\theta$ as a function of three different illumination parameters.

**Figure supplement 2.** Control for retinal origin of klinotaxis.

(corresponding to a negative subjective value), whereas its exploratory kinematics remain unchanged upon an increase of illumination (positive subjective value).

Two kinematic parameters can possibly impact the orientational diffusivity: the fraction of turning bouts $p_{turn}$ and their characteristic amplitude $\sigma_{turn}$. We thus extracted these two quantities and plotted them as a function of $\delta I/I$ (*Figure 3H–I*). They appear to equally contribute to the observed modulation.

To test whether this uniform phototactic process has a retinal origin, or whether it might be mediated by non-visual deep-brain photoreceptors (*Fernandes et al., 2012*), we ran similar assays on bi-enucleated fish. In this condition, no orientational bias was observed, which indicates that the retinal pathway is involved in orientational klinotaxis *Figure 3—figure supplement 2*, all p-values > 0.14, pairwise $T$-tests).

## A biased random walk model for phototaxis provides a quantitative prediction of light-driven orientational bias

In the preceding sections, we quantified how visual stimuli stochastically modulate specific kinematic parameters of the subsequent bout. We used these relationships to build a biased random walk model of phototaxis. We then tested how such a model could reproduce the statistical orientational biases toward the directions of minimal contrast and maximal illumination. The phototactic model thus incorporates a visually-driven bias to the discrete Markov-chain model introduced to represent the spontaneous navigation (*Figure 4A*). In line with the observation of *Figure 2M*, the rate of flipping orientational state (left-to-right or right-to-left) was a linear function of the imposed contrast: $k_{R \to L} = k_{flip} + ac$ and $k_{l \to r} = k_{flip} - ac$. The value of $a$ was set so as to capture the contrast-dependent orientational drift (*Figure 2G*) and was made dependent on bout index in order to account for the observed short-term habituation process (*Figure 2—figure supplement 2*).

The selection of bout type was in turn linearly modulated by the relative change in intensity after negative rectification, $[\delta I/I]^- = min(\delta I/I, 0)$. Hence, the turn-to-forward and forward-to-turn transition

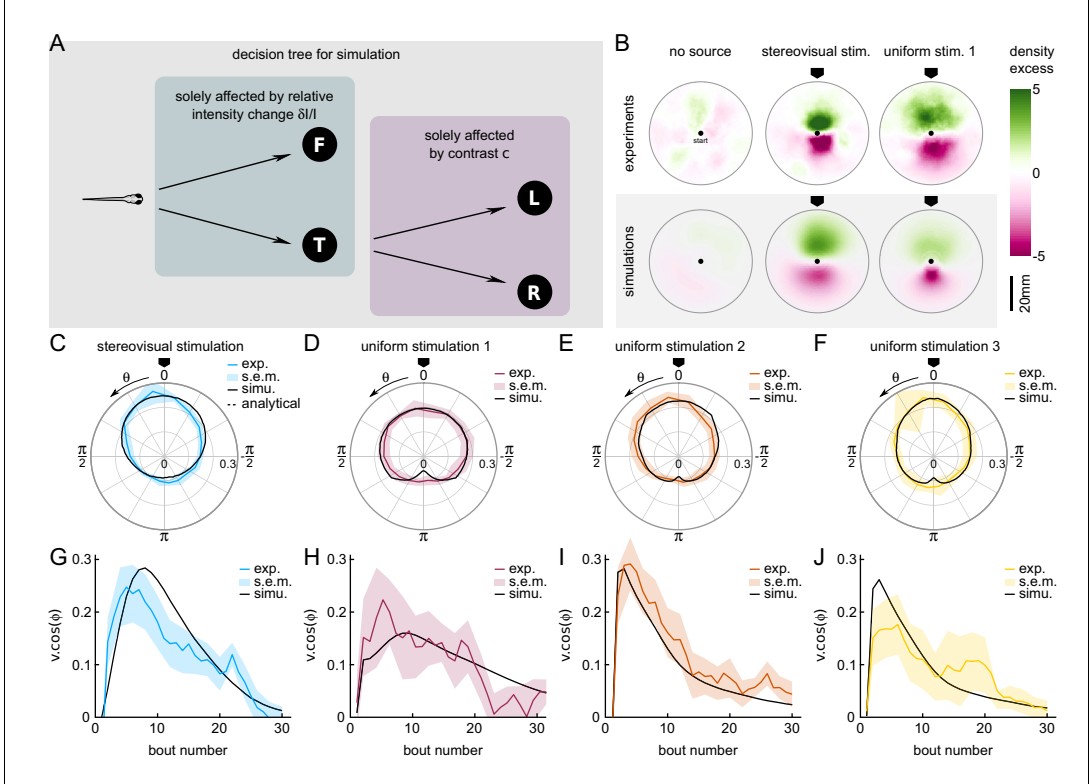

**Figure 4.** A Markov-chain model of phototaxis captures the observed orientational distribution. (**A**) Decision tree for simulation: selection of forward scoots vs turning bouts are governed by the relative intensity change at the previous bout. If a turning bout is triggered, the selection of left-right orientation is biased by the stereovisual contrast. (**B**) 2D density profiles computed from all experimental and simulated trajectories for the three different paradigms (no stimulation, lateralized illumination and uniform illumination). The color encodes the excess or deficit of density with respect to the radially-averaged density without any stimulation. (**C–F**) Experimental (color) and simulation (solid line) probability density distributions of orientations for the four phototactic configurations (stereo-visual stimulation, uniform stimulation with angular profiles 1 to 3). (**G–J**) Evolution of the projection of the resultant vector onto the direction of the light source as a function of the bout number for the experiment (color) and simulation (solid line). Error bar : standard error of the mean.

The online version of this article includes the following figure supplement(s) for figure 4:

**Figure supplement 1.** Inter-bout distance distribution.

rates read $k_{t \to f} = k_{turn} + \beta[\delta I/I]^-$ and $k_{f \to t} = k_{turn} - \beta[\delta I/I]^-$, respectively. We also imposed a linear modulation of the turn amplitude variance $\sigma_{turn} = \sigma_{turn}^{spont} - \gamma[\delta I/I]^-$. The values of $\beta$ and $\gamma$ were adjusted to reproduce the observed dependence of the turn-vs-forward ratio and bout amplitude with $\delta I/I$ (**Figure 3H–I**).

This stochastic model was tested under two conditions, tropo- and klino-phototaxis, similar to those probed in the experiments (**Figure 4B**). In order to account for the sampling bias associated with the finite size of the experimental ROI, the particles in the simulations also progressed in a 2D arena. At each time step, a forward displacement was drawn from a gamma distribution adjusted on the experimental data (**Figure 5—figure supplement 1**). Statistical analysis was restricted to bouts executed within a circular ROI as in the experimental assay.

The comparison of the data and numerical simulation is shown in **Figure 4C** for the tropotaxis protocol and in **Figure 4D–F** for the klinotaxis protocols. This minimal stochastic model quantitatively captures the distribution of orientations. It also reproduces the evolution of the orientational bias with the bout index as measured by the length of the resultant vector (**Figure 4G–J**).

# A neuronal model of the ARTR captures spontaneous and contrast-driven navigation

The behavioral description proposed above indicates that larvae navigation can be correctly accounted for by two independent stochastic processes: one that organizes the sequence of successive bouts amplitude and in particular the selection of forward *vs* turning events, while a second one selects the left *vs* right orientation of the turning bouts. These two processes are independently modulated by two distinct features of the visual stimulus, namely the global intensity changes and the stereo-visual contrast, leading to the two phototactic strategies.

This in turn suggests that, at the neuronal level, two independent circuits may control these characteristics of the executed swim bouts. As mentioned in the introduction, the ARTR is a natural candidate for the neuronal selection of bouts orientation. This small bilaterally-distributed circuit located in the anterior hindbrain displays antiphasic activity oscillation with ~ 20s period (*Ahrens et al., 2013*). The currently active region (left or right) constitutes a strong predictor of the orientation of turning bouts (*Dunn et al., 2016*). This circuit further integrates visual inputs as each ARTR subpopulation responds to the stimulation of the ipsilateral eye (*Wolf et al., 2017*).

Here, we adapted a minimal neuronal model of the ARTR, introduced in *Wolf et al. (2017)* to interpret the calcium recordings, and tested whether it could explain the observed statistics of exploration in both spontaneous and phototactic conditions. The architecture of the model is depicted in *Figure 5A* and the equations governing the network dynamics are provided in Appendix 2. The network consists of two modules selective for leftward and rightward turning, respectively. Recurrent excitation ($w_E$) drives self-sustained persistent activities over finite periods of time. Reciprocal inhibition ($w_I$) between the left and right modules endows the circuit with an antiphasic dynamics. Finally, each ARTR module receives an input current from the visual system proportional to the illumination of the ipsilateral eye. Such architecture gives rise to a stimulus-selective attractor as described in *Freeman (1995)* and *Wang (2002)*.

The various model parameters were adjusted in order to match the behavioral data (see Appendix 2). First, the self-excitatory and cross-inhibitory couplings were chosen such that the circuit displayed spontaneous oscillatory dynamics in the absence of sensory input. *Figure 5B* shows example traces of the two units' activity in this particular regime. From these two traces, we extracted a binary 'orientational state' signal by assigning to each time point a left or right value (indicated in red and blue, respectively), based on the identity of the module with the largest activity.

In the present approach, tail bouts are assumed to be triggered independently of the ARTR activity. The latter thus acts as mere orientational hub by selecting the orientation of the turning events: incoming bouts are oriented in the direction associated with the currently active module. In the absence of information regarding the circuit that organizes the swim bouts emission, their timing and absolute amplitude were drawn from the behavioral recordings of freely swimming larvae. Combined with the ARTR dynamics, this yielded a discrete sequence of simulated bouts (leftward, rightward and forward, *Figure 5B*, inset). With adequate choice of parameters, this model captures the orientational persistence mechanism as quantified by the slow decay of the turning bout autocorrelation with the interbout interval (*Figure 5C* and *Figure 5—figure supplement 1*).

In the presence of a lateralized visual stimulus, the oscillatory dynamics become biased towards the brighter direction (*Figure 5D–E*). Hence, illuminating the right eye favors longer periods of activation of the rightward-selective ARTR unit. The mean reorientation displays a quasi-linear dependence with the imposed contrast (*Figure 5D*) consistent with the behavioral observations (*Figure 2G*). At intermediate contrast values, the orientation of bouts remains stochastic; the effect of the contrast is to lengthen streaks of turning bouts toward the light (*Figure 5E*). We also tested whether this model could capture the competition mechanism between stereovisual bias and motor persistence, in both conflicting and reinforcement conditions. We thus computed the dependence of the flipping probability $p_{flip}$ as a function of the contrast in both conditions (*Figure 5F*). The resulting graph is in quantitative agreement with its experimental counterparts (*Figure 2M*).

We finally used this model to emulate a simulated phototactic task. In order to do so, a virtual fish was submitted to a contrast whose amplitude varied linearly with the animal orientation, as in the lateralized assay. When a turning bout was triggered, its orientation was set by the ARTR instantaneous activity while its amplitude was drawn from the experimental distributions. After a few bouts, a stationary distribution of orientation was reached that was biased toward the virtual light

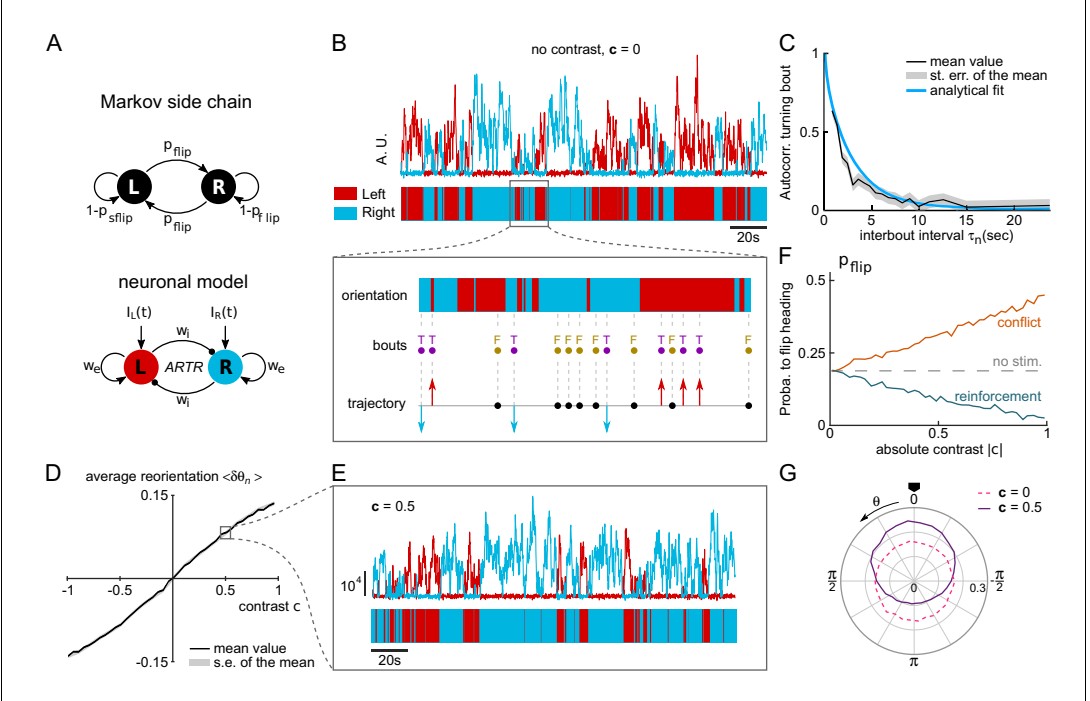

**Figure 5.** A neuronal model of turning bout selection captures spontaneous and contrast-driven navigation. (A) Scheme of the Markov-chain model of the orientation selection, and corresponding neuronal model of the ARTR. The latter consists of two units whose relative activation controls the orientation of bouts. Persistent and self-alternating dynamics result from the recurrent excitation ($w_E$) and reciprocal inhibition ($w_I$) between each unit. They further receive input currents proportional to the illumination of the ipsilateral eye. (B) Top: example traces of the simulated activity of the left (red) and right (blue) modules in the absence visual stimulation (AU : arbitrary units). These continuous dynamics control the alternation between right and left orientational states. Close-up: forward and turning bouts are triggered independently with a statistics drawn from the behavioral recordings. The orientational state governs the orientation of the turning bouts. (C) Orientation correlation of turning bouts (thresholded at 0.22 rad) as a function of the inter-bout interval $\tau_n$. Result from the neuronal model is in blue, experimental data are in black. (D) Mean reorientation $<\delta\theta>$ as a function of the contrast $c$. (E) Example traces of the simulated activity for a constant contrast $c = 0.5$. (F) Probability of flipping orientation as a function of the imposed contrast $c$ in situations of conflict or reinforcement (neuronal model). (G) Probability distribution function of $\theta$ for 10 simulated phototactic trajectories with a linear dependence of average reorientation on contrast. Each trajectory lasted 50,000 s. The dotted line is the orientational distribution in the absence of visual stimulation.

The online version of this article includes the following figure supplement(s) for figure 5:

**Figure supplement 1.** Comparison of experimental and simulated trajectories.
**Figure supplement 2.** Simulated trajectories with different inter-bout intervals $\tau_n$.

source (*Figure 5G*). Its profile was in quantitative agreement with its experimental counterpart (mean resultant vector length $v = 0.23$ in simulation for $v = 0.24$ in experimental data for bouts 2 to 17).

## Discussion

Sensorimotor transformation can be viewed as an operation of massive dimensionality reduction, in which a continuous stream of sensory and motor-related signals is converted into a discrete series of stereotyped motor actions. The challenge in understanding this process is (i) to correctly categorize the motor events, that is to reveal the correct parametrization of the motor repertoire, and (ii) to unveil the statistical rules for action selection. Testing the validity of such description can be done by building a minimal model based on these rules. If the model is correct, the motor variability unaccounted for by the model should be entirely random, that is independent of the sensorimotor history.

Here, we implemented a minimal model approach in order to unveil the basic rules underlying phototaxis. We showed that zebrafish light-driven orientational navigation can be quantitatively described by a stochastic model consisting of two independent Markov chains: one that selects

forward scoots vs turning bouts and a second one that sets the orientation of the latter. We established that the stereo-visual contrast and global intensity modulation act separately on each of these selection processes. The contrast induces a directed bias of turning bouts toward the illuminated side, but does not impact the prevalence of turning bouts vs forward scoots. Reversely, a global decrement in illumination increases the ratio of turning bouts but does not favor any particular direction. For the contrast-driven configuration (tropotaxis), the minimal model corresponds to an Ornstein-Uhlenbeck process (*Uhlenbeck and Ornstein, 1930*), which describes the dynamics of a diffusive brownian particle in a quadratic trap. In the klinotaxis configuration (in the absence of stereo-visual contrast), the orientational bias solely results from a light-dependent modulation of the diffusivity, a mechanism reminiscent of bacterial chemotaxis.

This stochastic minimal model is built on a simple decision tree (*Figure 4A*) with a set of binary choices. However, to fully capture the orientational dynamics, we had to incorporate the continuous increase in turning bout amplitude with the light decrement in an ad-hoc way. It is currently unclear whether all turn bouts in our experiments can be assigned to a single class of swim maneuvers that are modulated in amplitude, or whether these encompass distinct motor programs executed with varying frequencies. In the latter case, it might be possible to represent this amplitude modulation through an extension of the decision tree that would select between distinct turn bout categories.

Compared to previous studies on phototaxis, for example (*Burgess et al., 2010*), our approach allowed us to clearly disentangle the contributions of spatial (stereovisual contrast) and time-dependent (motion-induced change in global illumination) visual cues. Hence, the contrast-driven assays were performed under constant overall illumination intensity (the sum of left and right intensities). This allowed us to establish that, rather surprisingly, the probability of triggering a turn (vs a forward swim) is insensitive to the imposed contrast. This possibility constitutes an important asset with respect to standard experimental configurations, such as the one examined by *Burgess et al. (2010)*, in which the animal is submitted to an actual light source. Although these configurations provide a more realistic context, the visual stimulus effectively perceived by each eye can not be quantitatively assessed, which precludes the design of predictive models. Conversely, once adjusted on well-controlled virtual assays, our model could be numerically implemented in realistic environments, and the trajectories could then be directly confronted with behavioral data. This would require to first infer how the intensity impinging on each eye depends on the source distance and orientation relative to the animal body axis.

Another critical and distinct aspect of the present work is its focus on the steady-state dynamics. Our aim was to mimic the continuous exploration of an environment in which the brightness level displayed slowly varying angular modulations. The luminosity profiles were thus chosen to ensure that individual bouts elicited relatively mild changes in illumination. By doing so, we tried to mitigate visual startle responses that are known to be elicited upon sudden darkening (*Easter and Nicola, 1996*). Although we could not avoid the initial large drop in illumination at the onset of each trial, the associated short-term response (i.e. the first bout) was excluded from the analysis. In this respect, our experiment differs from the study of *Chen and Engert (2014)* in which a similar closed-loop setup was used to demonstrate the ability of larvae to confine their navigation within bright regions. This behavior was entirely controlled by the animal's response to light-on or light-off stimuli as it crossed the virtual border between a bright central disk and the dark outer area. These sharp transitions resulted in clear-cut behavioral changes that lasted for a few bouts. In comparison, our experiment addresses a different regime in which subtle light-driven biases in the spontaneous exploration cumulatively drive the animal toward brightest regions.

As we aimed to obtain a simple and tractable kinematic description, we ignored some other aspects of the navigation characteristics. First, we focused on the orientation of the animal and thus did not systematically investigate how the forward components of the swim bouts were impacted by visual stimuli. However, in the context of angle-dependent intensity profiles, this effect should not impact the observed orientational dynamics. More importantly, we ran most of our analysis using the bout number as a time-scale, and thus ignored possible light-driven modulations of the inter-bout intervals ($\tau_n$). We showed, however, that the orientational correlation is controlled by an actual time-scale. This result may have significant consequence on the fish exploration. In particular, we expect that changes in bout frequency, reflecting various levels of motor activity, may significantly affect the geometry of the trajectories (and not only the speed at which they are explored). We illustrated this process by running numerical experiments at similar flipping rate $k_{flip}$ but increasing bout

frequencies. The trajectories, shown in *Figure 5—figure supplement 2*, exhibit increasing complexity as measured by the fractal dimension. This mechanism may explain the changes in trajectories' geometry observed by *Horstick et al. (2017)* in response to sudden light dimming.

An important outcome of this study is to show that light-seeking navigation uses visual cues over relatively short time scales. The bouts statistics could be captured with a first-order autoregressive process, indicating that the stimulus perceived over one $\tau_n$ is sufficient to predict the forthcoming bout. However, one should be aware that such observation is only valid provided that the sensory context remains relatively stable. Hence for instance, a prolonged uniform drop in luminosity is known to enhance the overall motor activity (generally estimated by the average displacement over a period of time) for up to several tens of minutes (*Prober et al., 2006*; *Emran et al., 2007*; *Emran et al., 2010*; *Liu et al., 2015*). This long-term behavioral change, so-called photokinesis, might be regulated by deep brain photoreceptors (*Fernandes et al., 2012*; *Horstick et al., 2017*) and thus constitutes a distinct mechanism. One particularly exciting prospect will be to understand how such behavioral plasticity may not only modulate the spontaneous activity (*Johnson et al., 2019*) but also affects the phototactic dynamics.

One of the motivations of minimal behavioral models is to facilitate the functional identification and modeling of neural circuits that implement the identified sensorimotor operations in the brain. Here, we used the behavioral results to propose a neuronal model of the ARTR that quantitatively reproduces non-trivial aspects of the bout selection process. This recurrent neural circuit is a simplified version of working memory models developed by *Brunel and Wang (2001)*; *Wang (2001)*; *Wang (2002)*; *Wang (2008)* and adapted in *Wang (2002)* for a decision task executed in the parietal cortex (*Shadlen and Newsome, 1996*; *Shadlen and Newsome, 2001*). In this class of models, the binary decision process reflects the competition between two cross-inhibitory neural populations. The circuit is endowed with two major functional capacities: (1) it can maintain mnemonic persistent activity over long periods of time, thanks to recurrent excitatory inputs; (2) it can integrate sensory signals in a graded fashion to continuously bias the statistics of the decision. This model thus naturally recapitulates the major functional features of the sensory-biased Markov side-chain - motor persistence and contrast-driven continuous bias - that organizes the orientation selection.

It is tempting to generalize about this behavior-to-circuit approach, at least in small animals such as Zebrafish or *Drosophila*, by representing any behavior as a coordinated sequence of competing elemental actions biased by sensory feedback and organized within a hierarchical decision tree. The identification of such decision trees through quantitative behavioral analysis may provide a blueprint of the brain functional organization and significantly ease the development of circuit models of brain-scale sensorimotor computation.

## Materials and methods

### Zebrafish maintenance and behavioral setup

All experiments were performed on wild-type Zebrafish (Danio Rerio) larvae aged 5 to 8 days post-fertilization. Larvae were reared in Petri dishes in E3 solution on a 14/10 hr light/dark cycle at 28°C, and were fed powdered nursery food every day from 6 dpf.

Experiments were conducted during daytime hours (10 am to 6 pm). The arena consisted of a 14 cm in diameter Petri dish containing E3 medium. It was placed on a screen illuminated from below by a projector (ASUS S1). Infrared illumination was provided by LEDs to enable video-monitoring and subsequent tracking of the fish. We used an IR-sensitive Flea3 USB3 camera (FL3-U3-13Y13M-C, Point Grey Research, Richmond, BC, Canada) with an adjustable macro lens (Zoom 7000, Navitar, USA) equipped with an IR filter. The experimental setup was enclosed in a light-tight rig, which was maintained at 26°C using 'The Cube' (Life Imaging Services).

For the stereovisual paradigm $N = 47$ larvae were tested, and $N = 37$ for the temporal paradigm [(uniform 1) : 12, (uniform 2) : 11, (uniform 3) : 14]. All fish ($N=75$) that navigated in the ROI for a significant period of time during the habituation period were also used to assess spontaneous navigation statistics.

## Behavioral assay

Closed-loop tracking and visual stimulation were performed at a mean frequency of 35 Hz, with a custom-written software (*Karpenko, 2019b*; copy archived at https://github.com/elifesciences-publications/Analysis_Behavioral_Phototaxis) in Matlab (The MathWorks), using the PsychToolBox (PTB) version 3.0.14 add-on. Positions and orientations (heading direction) of the fish, as well as bouts characteristics, were extracted online and the illumination pattern was updated accordingly, with a maximum latency of 34 ms. Heading direction was extracted with an accuracy of + /- 0.05 rad ($\sim 3°$). Behavioral monitoring was restricted to a circular central region of interest (ROI) of 8.2 cm diameter. When outside the ROI, the fish was actively brought back into the ROI through the opto-motor reflex (OMR), using a concentrically moving circular pattern. One second after the fish re-entered the ROI, a new recording sequence was started.

Prior to the phototactic assay, all tested fish were subjected to a period of at least 8 min of habituation under whole-field illumination at an intensity of $I_{max} = 450 \mu W.cm^{-2}$. For both phototactic paradigms, the absolute orientation of the virtual source was randomly selected when initiating a new experimental sequence (each time the animal would re-enter the ROI). The orientation of the fish relative to the light source $\theta_n$ was calculated online using the absolute orientation of the fish $\alpha_n$ and the orientation of the virtual light source $\alpha_{source} : \theta_n = \alpha_n - \alpha_{source}$.

*Lateralized paradigm.* A circle of 6 cm in diameter was projected under and centered on the fish. The circle was divided into two parts, covering the left and right side of the fish. The separation between the two parts corresponded to the animal's midline. A separation band (2 mm thick) and an angular sector ($30°$) in front of the animal were darkened to avoid interception of light coming from the right side of the fish by its left eye and *vice-versa*. The left and right intensities ($I_L$ and $I_R$) were varied linearly as a function of $\theta$, such that $I_L + I_R = I_{max}$. Since during the habituation period, the whole arena was lit at maximum intensity $I_{max}$, the total intensity received by the fish drops by a factor of $\approx 2$ with the establishment of the circle, at the onset of the assay.

Although our imposed contrast profile displays two angles for which the contrast is null, namely $\theta = 0$ and $\pi$, only does the first one correspond to a stable equilibrium point. When $\theta$ is close to zero, any excursion away from this particular direction results in a contrast that drives the animal back to the null angle. Conversely, when $\theta \approx \pi$, the contrast drives the animal away from $\pi$ (unstable equilibrium).

*Temporal paradigm.* The whole arena was illuminated with an intensity locked onto the fish orientation $\theta$ relative to a virtual light source. The initial orientation was randomly chosen at the beginning of a recording sequence. Three different intensity angular profiles were implemented: (uniform 1) a sinusoidal profile, with a maximum intensity of 60% of $I_{max}$, (uniform 2) an exponential profile, with a maximum intensity of 60% of $I_{max}$ and finally (uniform 3) an exponential profile with a maximum intensity of 30% of $I_{max}$.

## Data analysis

Data analysis was performed using a custom-written code in Matlab. All analysis programs and data are available at *Karpenko (2019a)*.

When representing the mean of one variable against another, bin edges were chosen such that each bin would encompass the same number of data points. Circular statistics analyses (mean, variance, uniformity) and circular statistics tests, namely the circular V-test of non-uniformity of data and the one-sample test for the mean angle of a circular distribution (tested on the orientation of the light virtual light source) were performed using CircStat toolbox for Matlab (*Berens, 2009*).

Individual fish often exhibit a small yet consistent bias toward one direction (either leftward or rightward). This bias was subtracted before performing the different analyses, in order to guarantee that $<\alpha> = 0$ in the absence of a stimulus. The distribution of reorientation angles $\delta\theta_n$ during spontaneous swimming periods was fitted with a constrained double-Gaussian function. We imposed that both the mean absolute angle and variance of the fitting function be consistent with the experimental measurements. This yields an expression with only one independent fitting parameter $p_{turn}$ in the form:

$$f(x) = \frac{1}{\sqrt{2\pi}} \left( \frac{p_{turn}}{\sigma_{turn}} e^{-\frac{1}{2}\left(\frac{x}{\sigma_{turn}}\right)^2} + \frac{1-p_{turn}}{\sigma_f} e^{-\frac{1}{2}\left(\frac{x}{\sigma_{fwd}}\right)^2} \right) \qquad (1)$$

using

$$\sigma_{turn} = \frac{\mu_{abs}p_{turn} + \sqrt{\mu_{abs}^2 p_{turn}^2 - \left[\mu_{abs}^2 - V(1-p_{turn})\right]\left[p_{turn}^2 + p_{turn}(1-p_{turn})\right]}}{p_{turn}^2 + p_{turn}(1-p_{turn})}$$

and

$$\sigma_{fwd} = \frac{\mu_{abs}\sqrt{\pi/2} - p_{turn}\sigma_{turn}}{1 - p_{turn}}$$

with

$$\mu_{abs} = \langle|\delta\theta|\rangle, V = \langle\delta\theta^2\rangle$$

To evaluate the mean and variance of the forward and turn bouts under various visual contexts, the distributions in different bins were also fitted with a constrained double-Gaussian model as in (1). The stereovisual data distributions were fitted with two additional mean terms $\mu_{turn}$ and $\mu_{fwd}$ ; and for the klinotaxis assay, with a constraint on $\sigma_{fwd}$ and $\mu_{fwd}$. The bins were constructed either on the contrast $c$ experienced just before bout $n$ or on the relative difference of intensity experienced at bout $n-1 : \delta I/I = 2\frac{I_{n-1}-I_{n-2}}{I_{n-1}+I_{n-2}}$.

All distributions of $\theta_n$ and analyses of bias were computed using trajectories from bout index two to the median number of bouts per sequence in each type of experiment. The median number of bouts in each experiment was $med_{stereo} = 17$ for the tropotaxis experiment, and 27, 15, 17 for the klinotaxis assays for the 3 profiles 1–3, respectively.

## Numerical simulations

The Markov-chain model simulations were performed using a custom-written code in MATLAB (*Karpenko, 2019b*). Initial orientations and positions within the ROI were randomly sampled from, respectively, a uniform distribution and a normal distribution centered on a circle of radius 20 mm from the center of the ROI with a standard deviation of 1.3 mm (mimicking the starting points of experimental data).

At each step, an angular step-size is drawn from the data: either from the turning distribution with a probability $p_{turn}$ or from the forward distribution with a probability $1 - p_{turn}$. Respective means are $\mu_{turn}$ and $\mu_{fwd}$ and standard deviation $\sigma_{turn}$ and $\sigma_{fwd}$. The left-vs-right orientations of the turns is set by the probability of flipping sides $p_{flip}$. For the spatially constrained simulations, the walker also draws a distance step-size (between two successive positions) from two different gamma distributions: one for the turning bouts, a second one for the scoots. Under neutral conditions (uniform illumination), all parameters are constant.

For the simulation under stereovisual phototactic conditions, $p_{flip}$ was varied linearly with the contrast (based on the data represented in *Figure 2M*). When simulating temporal phototaxis, the parameters $\sigma_{turn}$ and $p_{turn}$ were modulated by the relative illumination change $\delta I/I$ experienced at the previous steps (as represented in *Figure 3H–I*).

## Acknowledgements

We thank the IBPS fish facility staff, and in particular Alex Bois and Stéphane Tronche, for their help with the fish maintenance. We are grateful to Raphaël Voituriez for fruitful discussions on the theoretical aspects. This work was funded by the CNRS and Sorbonne University, and supported in part by the National Science Foundation under Grant No. NSF PHY-1748958.

## Additional information

### Funding

| Funder | Grant reference number | Author |
|---|---|---|
| Human Frontier Science Program | RGP0060/2017 | Georges Debrégeas |
| H2020 European Research Council | 71598 | Volker Bormuth |
| Agence Nationale de la Recherche | ANR-16-CE16-0017 | Raphaël Candelier<br>Georges Debrégeas |
| Fondation pour la Recherche Médicale | FDT201904008219 | Sophia Karpenko |
| Inserm | ATIP-Avenir program | Volker Bormuth |
| Fondation pour la Recherche Médicale | SPF201809007064 | Sebastien Wolf |

The funders had no role in study design, data collection and interpretation, or the decision to submit the work for publication.

### Author contributions
Sophia Karpenko, Conceptualization, Data curation, Formal analysis, Investigation; Sebastien Wolf, Formal analysis, Investigation; Julie Lafaye, Data curation, Investigation; Guillaume Le Goc, Data curation, Methodology; Thomas Panier, Investigation, Methodology; Volker Bormuth, Conceptualization, Supervision, Methodology; Raphaël Candelier, Conceptualization, Data curation, Supervision, Visualization, Methodology; Georges Debrégeas, Conceptualization, Formal analysis, Supervision

### Author ORCIDs
Raphaël Candelier (ID) http://orcid.org/0000-0002-1523-6249
Georges Debrégeas (ID) https://orcid.org/0000-0003-3698-4497

### Ethics
Animal experimentation: All experiments were approved by Le Comité d'Éthique pour l'Expérimentation Animale Charles Darwin C2EA-05 (02601.01).

### Decision letter and Author response
Decision letter https://doi.org/10.7554/eLife.52882.sa1
Author response https://doi.org/10.7554/eLife.52882.sa2

## Additional files
### Supplementary files
- Transparent reporting form

### Data availability
Data and analysis codes are available at Dryad Digital Repository, DOI: https://doi.org/10.5061/dryad.v9s4mw6qx.

The following dataset was generated:

| Author(s) | Year | Dataset title | Dataset URL | Database and Identifier |
|---|---|---|---|---|
| Debregeas G, Karpenko S, Wolf S, Lafaye J, Le Goc G, Panier T, Candelier R, Bormuth V | 2019 | From behavior to circuit modeling of light-seeking navigation in Zebrafish larvae | http://doi.org/10.5061/dryad.v9s4mw6qx | Dryad Digital Repository, 10.5061/dryad.v9s4mw6qx |

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

## Appendix 1

# Modelling spontaneous navigation of zebrafish larvae

We model the discrete trajectories (sequences of bouts) as a stochastic process using two independent Markov chains, depicted in *Figure 1E*. The *bout type* chain (top) controls the alternation between forward and turning bouts, with possible states $F_n$ and $T_n$ at time $n$, while the *side* chain (bottom) controls the left/right orientations of turning bouts, with possible states $L_n$ and $R_n$. All possible states are combinations of the states of the two chains, namely $\{FL\}_n$, $\{FR\}_n$, $\{TL\}_n$, and $\{TR_n\}$. The transition rates of the bout type chain are $k_{F \to T}$ and $k_{T \to F}$, where $k_{F \to T}/k_{T \to F} = p_{turn}$ is the overall fraction of turning bouts. For the side chain, under constant uniform illumination, the right and left states are equiprobable, and the two transition probabilities are thus equal: $k_{R \to L} = k_{L \to R} = p_{flip}$.

The two chains operate synchronously: at every time step transitions on both chains are triggered simultaneously, and a reorientation value $\delta\alpha_n$ is drawn based on the resulting state. When the fish is in a turning state, $\{TL\}_n$ or $\{TR\}_n$, the reorientation angle is sampled from the positive and negative side of a centered normal distribution with standard deviation $\sigma_{turn}$ for left and right turns, respectively. When the fish is in a forward state, $\{FL\}_n$ or $\{FR\}_n$, the reorientation angle is drawn from a normal distribution with standard deviation $\sigma_{fwd}$. Therefore, for forward bouts the resulting $\delta\alpha_n$ can be positive or negative, irrespective of the left/right state of the side chain. Altogether, the general statistical distribution of turning amplitudes $\delta\alpha_n$ used in *Figure 1F* reads

$$P(\delta\alpha_n) = \phi_t(\delta\alpha_n) + \phi_f(\delta\alpha_n) \tag{A1}$$

with

$$\phi_t = p_{turn}\mathcal{N}(0, \sigma_{turn}^2) \quad \text{and} \quad \phi_f = (1 - p_{turn})\mathcal{N}(0, \sigma_{fwd}^2) \tag{A2}$$

# Mean amplitude at $n + 1$

Within this framework, one can analytically compute the mean square angle at time $n + 1$, as detailed below:

$$\langle \delta\alpha_{n+1}^2 \rangle = \mathbb{P}(F_{n+1})\sigma_{fwd}^2 + \mathbb{P}(T_{n+1})\sigma_{turn}^2 \tag{A3}$$

The two probabilities read:

$$\mathbb{P}(F_{n+1}) = \mathbb{P}(F_n)(1 - k_{F \to T}) + \mathbb{P}(T_n)k_{T \to F}$$

$$\mathbb{P}(T_{n+1}) = \mathbb{P}(T_n)(1 - k_{T \to F}) + \mathbb{P}(F_n)k_{F \to T}$$

such that

$$\langle \delta\alpha_{n+1}^2 \rangle = \mathbb{P}(F_n)\left[\sigma_{fwd}^2 + k_{F \to T}(\sigma_{turn}^2 - \sigma_{fwd}^2)\right] + \mathbb{P}(T_n)\left[\sigma_{turn}^2 + k_{T \to F}(\sigma_{fwd}^2 - \sigma_{turn}^2)\right]$$

Using the functions defined in *Equation A2* we introduce the function $f(\delta\alpha_n)$:

$$f(\delta\alpha_n) = \mathbb{P}(T_n|\delta\alpha_n) = \frac{\phi_t(\delta\alpha_n)}{\phi_t(\delta\alpha_n) + \phi_f(\delta\alpha_n)} \tag{A4}$$

The mean square amplitude at time $n + 1$ can thus be written, as a function of $\delta\alpha_n$ as:

$$\langle \delta\alpha_{n+1}^2 \rangle = \sigma_{fwd}^2 + k_{F \to T}(\sigma_{turn}^2 - \sigma_{fwd}^2) + f(|\delta\alpha_n|)(\sigma_{turn}^2 - \sigma_{fwd}^2)(1 - k_{T \to F} - k_{F \to T}) \tag{A5}$$

This expression is used to fit the data in *Figure 1F* and estimate the two transition rates. These are found to be close to the ratio of turning and forward bouts, that is $k_{F \to T}/p_{turn} = k_{T \to F}/(1 - pturn) \approx 0.8$. In the following, we set $k_{F \to T} = p_{turn}$ and

$k_{T \to F} = 1 - p_{turn}$, thus ignoring the weak memory component in the selection of turning vs forward bouts.

## Mean reorientation at $n + 1$

Similarly, one can compute the theoretical expression of the mean reorientation angle at time $n + 1$:

$$\langle \delta \alpha_{n+1} \rangle = \mathbb{P}(\{FL\}_{n+1})\mu_f + \mathbb{P}(\{FR\}_{n+1})\mu_f + \mathbb{P}(\{TL\}_{n+1})\mu_L + \mathbb{P}(\{TR\}_{n+1})\mu_R \tag{A6}$$

with

$$\mu_f = 0 \quad \text{and} \quad \mu_L = -\mu_R = \sqrt{\frac{2}{\pi}}\sigma_{turn} \tag{A7}$$

Then:

$$\mathbb{P}(\{TL\}_{n+1}) = \mathbb{P}(T_{n+1})\mathbb{P}(L_{n+1}) = p_{turn}\left[p_{flip}\mathbb{P}(R_n) + (1-p_{flip})\mathbb{P}(L_n)\right]$$

$$\mathbb{P}(\{TR\}_{n+1}) = \mathbb{P}(T_{n+1})\mathbb{P}(R_{n+1}) = p_{turn}\left[p_{flip}\mathbb{P}(L_n) + (1-p_{flip})\mathbb{P}(R_n)\right]$$

and

$$\langle \delta \alpha_{n+1} \rangle = p_{turn}(1 - 2p_{flip})\sqrt{\frac{2}{\pi}}\sigma_{turn}[\mathbb{P}(L_n) - \mathbb{P}(R_n)] \tag{A8}$$

Without further assumption, this simply confirms $\langle \delta \alpha_{n+1} \rangle = 0$. Given the reorientation at time $n$, this expression now writes:

$$\langle \delta \alpha_{n+1} \rangle_{\delta \alpha_n} = p_{turn}(1 - 2p_{flip})\sqrt{\frac{2}{\pi}}\sigma_{turn}[\mathbb{P}(L_n|\delta \alpha_n) - \mathbb{P}(R_n|\delta \alpha_n)] \tag{A9}$$

Since

$$\mathbb{P}(L_n|\delta \alpha_n) = \mathbb{P}(L_n|T_n, \delta \alpha_n)\mathbb{P}(T_n|\delta \alpha_n) + \mathbb{P}(L_n|F_n, \delta \alpha_n)\mathbb{P}(F_n|\delta \alpha_n)$$

$$\mathbb{P}(R_n|\delta \alpha_n) = \mathbb{P}(R_n|T_n, \delta \alpha_n)\mathbb{P}(T_n|\delta \alpha_n) + \mathbb{P}(R_n|F_n, \delta \alpha_n)\mathbb{P}(F_n|\delta \alpha_n)$$

and

$$\mathbb{P}(L_n|F_n, \delta \alpha_n) = \mathbb{P}(R_n|F_n, \delta \alpha_n) = 1/2$$

we obtain

$$\langle \delta \alpha_{n+1} \rangle_{\delta \alpha_n} = [\mathbb{P}(L_n|T_n, \delta \alpha_n) - \mathbb{P}(R_n|T_n, \delta \alpha_n)]p_{turn}(1 - 2p_{flip})\sqrt{\frac{2}{\pi}}\sigma_{turn}f(\delta \alpha_n) \tag{A10}$$

Then, noting that

$$\begin{cases} \mathbb{P}(L_n|T_n, \delta \alpha_n > 0) = 1 \\ \mathbb{P}(R_n|T_n, \delta \alpha_n > 0) = 0 \end{cases} \quad \text{and} \quad \begin{cases} \mathbb{P}(L_n|T_n, \delta \alpha_n < 0) = 0 \\ \mathbb{P}(R_n|T_n, \delta \alpha_n < 0) = 1 \end{cases}$$

we finally obtain the formula used to fit the data in **Figure 1G**:

$$\langle \delta \alpha_{n+1} \rangle_{\delta \alpha_n} = sign(\delta \alpha_n)\sqrt{\frac{2}{\pi}}p_{turn}(1 - 2p_{flip})\sigma_{turn}f(\delta \alpha_n) \tag{A11}$$

## Autocorrelation of the reorientations

One can then compute the correlation of reorientation amplitudes, defined for $q \in \mathbb{N}^*$ as:

$$C_q = \frac{\langle \delta\alpha_n \delta\alpha_{n+q} \rangle - \langle \delta\alpha_n \rangle \langle \delta\alpha_{n+q} \rangle}{\sqrt{\langle \delta\alpha_n^2 \rangle}\sqrt{\langle \delta\alpha_{n+q}^2 \rangle}} = \frac{\langle \delta\alpha_n \delta\alpha_{n+q} \rangle}{\langle \delta\alpha_n^2 \rangle} \tag{A12}$$

with the normalization coefficient equal to the variance of reorientations

$$\langle \delta\alpha_n^2 \rangle = p_{turn}\sigma_{turn}^2 + (1 - p_{turn})\sigma_{fwd}^2 \tag{A13}$$

The term $\langle \delta\alpha_n \delta\alpha_{n+q} \rangle$ can be computed in a similar manner as for **Equation A6**, but with more terms corresponding to the 16 possible combinations of states:

$$\{FL\}_n, \{FL\}_{n+q} \quad \{FL\}_n, \{FR\}_{n+q} \quad \{FL\}_n, \{TL\}_{n+q} \quad \{FL\}_n, \{TR\}_{n+q}$$

$$\{FR\}_n, \{FL\}_{n+q} \quad \{FR\}_n, \{FR\}_{n+q} \quad \{FR\}_n, \{TL\}_{n+q} \quad \{FR\}_n, \{TR\}_{n+q}$$

$$\{TL\}_n, \{FL\}_{n+q} \quad \{TL\}_n, \{FR\}_{n+q} \quad \{TL\}_n, \{TL\}_{n+q} \quad \{TL\}_n, \{TR\}_{n+q}$$

$$\{TR\}_n, \{FL\}_{n+q} \quad \{TR\}_n, \{FR\}_{n+q} \quad \{TR\}_n, \{TL\}_{n+q} \quad \{TR\}_n, \{TR\}_{n+q}$$

Only the four states in the bottom-right corner have a finite contribution, since all the others terms are multiplied by $\mu_f = 0$. Thus:

$$\langle \delta\alpha_n \delta\alpha_{n+q} \rangle = \mathbb{P}(\{TL\}_n, \{TL\}_{n+q})\mu_L^2 + \mathbb{P}(\{TL\}_n, \{TR\}_{n+q})\mu_L\mu_R +$$
$$\mathbb{P}(\{TR\}_n, \{TL\}_{n+q})\mu_R\mu_L + \mathbb{P}(\{TR\}_n, \{TR\}_{n+q})\mu_R^2$$

and, using **Equation A7** and

$$\mathbb{P}(\{TL\}_n \{TL\}_{n+q}) = \mathbb{P}(T_n)\mathbb{P}(T_{n+q})\mathbb{P}(L_n)\mathbb{P}(L_{n+q}|L_n) = \frac{p_{turn}^2}{2}\mathbb{P}(L_{n+q}|L_n)$$

$$\mathbb{P}(\{TL\}_n \{TR\}_{n+q}) = \mathbb{P}(T_n)\mathbb{P}(T_{n+q})\mathbb{P}(L_n)\mathbb{P}(R_{n+q}|L_n) = \frac{p_{turn}^2}{2}\mathbb{P}(R_{n+q}|L_n)$$

$$\mathbb{P}(\{TR\}_n \{TL\}_{n+q}) = \mathbb{P}(T_n)\mathbb{P}(T_{n+q})\mathbb{P}(R_n)\mathbb{P}(L_{n+q}|R_n) = \frac{p_{turn}^2}{2}\mathbb{P}(L_{n+q}|R_n)$$

$$\mathbb{P}(\{TR\}_n \{TR\}_{n+q}) = \mathbb{P}(T_n)\mathbb{P}(T_{n+q})\mathbb{P}(R_n)\mathbb{P}(R_{n+q}|R_n) = \frac{p_{turn}^2}{2}\mathbb{P}(R_{n+q}|R_n)$$

and noting that

$$\mathbb{P}(L_{n+q}|L_n) = \mathbb{P}(R_{n+q}|R_n) = \sum_{\substack{i=1 \\ i\sim\text{odd}}}^{q+1} \binom{q}{i} p_{flip}^{i-1}(1-p_{flip})^{q-i+1}$$

$$\mathbb{P}(L_{n+q}|R_n) = \mathbb{P}(R_{n+q}|L_n) = \sum_{\substack{i=1 \\ i\sim\text{even}}}^{q+1} \binom{q}{i} p_{flip}^{i-1}(1-p_{flip})^{q-i+1}$$

one obtains

$$\langle \delta\alpha_n \delta\alpha_{n+q} \rangle = p_{turn}^2 \left[ \sum_{i=1}^{q+1} (-1)^{q+1} q i p_{flip}^{i-1}(1-p_{flip})^{q-i+1} \right] \mu_L^2 = \frac{2}{\pi}(1-2p_{flip})^q p_{turn}^2 \sigma_{turn}^2$$

and finally:

$$C_q = \frac{2}{\pi} \frac{p_{turn}^2 \sigma_{turn}^2}{p_{turn}\sigma_{turn}^2 + (1-p_{turn})\sigma_{fwd}^2}(1-2p_{flip})^q \tag{A14}$$

This is the equation used to fit the data in **Figure 1H**.

An estimate of $p_{flip}$ was calculated as follows. If only turns are considered :

$$\langle \delta\alpha_n \delta\alpha_{n+1} \rangle = p_{turn}(1-p_{flip})\langle |\delta\alpha_n||\delta\alpha_{n+1}| \rangle - p_{turn}p_{flip}\langle |\delta\alpha_n||\delta\alpha_{n+1}| \rangle$$
$$= p_{turn}^2[(1-2p_{flip})\langle |\delta\alpha_n||\delta\alpha_{n+1}| \rangle]$$

thus :

$$\frac{\langle \delta\alpha_n \delta\alpha_{n+1} \rangle}{\langle |\delta\alpha_n||\delta\alpha_{n+1}| \rangle} = p_{turn}^2(1-2p_{flip}) = C_1$$

And finally

$$p_{flip} = \frac{1}{2}\left(1 - \frac{C_1}{p_{turn}^2}\right) \tag{A15}$$

## Mean square reorientation (MSR)

The mean square reorientation for a lag $q \in \mathbb{N}^*$ is defined by:

$$M_q = \left\langle \left(\alpha_{n+q} - \alpha_n\right)^2 \right\rangle \tag{A16}$$

and can be expressed as a sum of correlations as follows:

$$
\begin{aligned}
M_q &= \left\langle \left(\sum_{i=1}^{q} \delta\alpha_{n+i-1}\right)^2 \right\rangle \\
&= \left\langle \sum_{i=1}^{q}\sum_{j=1}^{q} \delta\alpha_{n+i-1}\delta\alpha_{n+j-1} \right\rangle \\
&= \sum_{i=1}^{q}\sum_{j=1}^{q} \left\langle \delta\alpha_{n+i-1}\delta\alpha_{n+j-1} \right\rangle \\
&= q\langle \delta\alpha_n^2 \rangle + \sum_{i=1}^{q}\sum_{\substack{j=1 \\ j \neq i}}^{q} \left\langle \delta\alpha_{n+i-1}\delta\alpha_{n+j-1} \right\rangle \\
&= q\langle \delta\alpha_n^2 \rangle + 2\sum_{i=1}^{q-1}(q-i)\langle \delta\alpha_n\delta\alpha_{n+i} \rangle \\
&= \left[q + 2\sum_{i=1}^{q-1}(q-i)C_i\right]\langle \delta\alpha_n^2 \rangle
\end{aligned}
$$

and using *Equation A13* we finally obtain the expression used in *Figure 1I*:

$$M_q = \left[q + 2\sum_{i=1}^{q-1}(q-i)C_i\right]\left(p_{turn}\sigma_{turn}^2 + (1-p_{turn})\sigma_{fwd}^2\right) \tag{A17}$$

## Appendix 2

### Neuronal model of the ARTR

The architecture of the ARTR neuronal model is shown in *Figure 5A*. The circuit consists of two modules selective to lefward or rightward turning. Each module receives recurrent excitatory, cross-inhibitory and sensory inputs. The firing rates of the left/right ARTR modules, noted $r_{L,R}$, are governed by two differential equations:

$$\begin{cases} \tau \dot{r}_L = -r_L + \phi(w_E r_L - w_I r_R + I_0 + I_L(t)) + \epsilon(t) \\ \tau \dot{r}_R = -r_R + \phi(w_E r_R - w_I r_L + I_0 + I_R(t)) + \epsilon(t) \end{cases} \tag{A18}$$

where $w_E r_{L,R}$ is the recurrent excitatory current and $w_I r_{L,R}$ is the cross-inhibitory current originating from the contralateral side of the network. $I_0$ is a constant input current and $\epsilon(t)$ is a white noise. The function $\phi$ is a non-negative spiking constraint such that $\phi(x>0) = x$ and $\phi(x<0) = 0$. We fixed $\tau = 100ms$, a typical slow synaptic time constant, as in *Wang (2002)*. The constant input current is set to $I_0 = 20s^{-1}$ and the standard deviation of the noise current $\epsilon$ is set at $500s^{-1}$ as in *Wolf et al. (2017)*. $I_R$ and $I_L$ are the visual input currents, proportional to the intensity impinging the right and left eyes, respectively.

In this dynamical system, $w_I$ controls the anticorrelation between left and right module activities. We fixed $w_I = 7$ such that the anticorrelation of the left and right signals (in the absence of visual inputs) was comparable to the value -0.4 measured through calcium imaging of the ARTR as reported in *Wolf et al. (2017)*. The parameter $w_E$ controls the ability for each side of the network to exhibit stable activity across time periods longer than $\tau$. The network exhibits three different dynamic regimes depending on $w_E$. One is characterized by an absence of stable activity (low $w_E$). For $w_E \sim 1$, one module is constantly active while the other remains silent. At intermediate values of $w_E$, the network displays stochastic slow alternations between both states. The fixation time, that is the characteristic decay time of the autocorrelation of $r_{L,R}$, is governed by $w_E$. We chose $w_E = 0.925$ such that the auto-correlation in orientation of turning bouts is similar to its experimental counterpart (*Figure 5C* and *Figure 1J*).

To examine the effect of a stereovisual contrast $c$ on the network dynamics, the sensory input currents were set such that:

$$\begin{cases} I_L(t) = I_{light}(1-c)/2 \\ I_R(t) = I_{light}(1+c)/2 \end{cases} \tag{A19}$$

The value of the maximum current $I_{light}$ was set at $I_{light} = 1000s^{-1}$ in order to reproduce the contrast-dependent orientational bias (*Figure 5D* and *Figure 2G*).

