## [Decision Letter]

**Acceptance summary:**

In this manuscript, Karpenko and colleagues dissect the sensorimotor rules directing positive phototaxis in zebrafish larvae, combining virtual reality experiments with computational modeling. By evaluating how the detections of spatial and temporal differences in stimulus intensity contribute to motor responses, the behavioral experiments and analysis significantly advances the study of the navigation algorithms directing fish phototaxis. In particular, the authors demonstrate that reorientation modulations (of both the turn amplitude and the probability of turning) depend on the relative changes in stimulus intensity that the animal experiences. In addition, they show that turns are directed by a competition between two mechanisms: (i) a stereo-visual bias and (ii) motor persistence. To determine whether a minimal stochastic model is sufficient to account for the overall navigation of real fish, the authors use simulations to reproduce the sensory modulation of transitions between behavioral states as well as orientational biases observed in real fish. Going beyond the "post-diction" of behavioral trends, the authors take advantage of modeling to explore a potential neural-circuit mechanism underlying action selection in the hindbrain. These results create a roadmap to guide similar analyses of orientation behaviors to other sensory modalities in fish and in other animals, providing a new framework for the study of navigation and foraging and the neural processing underlying it.

**Decision letter after peer review:**

Thank you for submitting your article "From behavior to circuit modeling of light-seeking navigation in Zebrafish larvae" for consideration by *eLife*. Your article has been reviewed by two peer reviewers, including Gordon J Berman as the Reviewing Editor and Reviewer #1, and the evaluation has been overseen by Ronald Calabrese as the Senior Editor.

The reviewers have discussed the reviews with one another and the Reviewing Editor has drafted this decision to help you prepare a revised submission.

Summary:

In this manuscript, Karpenko and colleagues dissect the sensorimotor rules directing positive phototaxis in zebrafish larvae. To this end, they combine a cutting-edge assay featuring virtual realities with computational modeling. The behavioral analysis is based on a thorough statistical analysis of the kinematic variables underpinning the navigation of free moving fishes. This analysis creates a roadmap to guide similar analysis of orientation behaviors to other sensory modalities in fish and in other animals. By evaluating how the detections of spatial and temporal differences in stimulus intensity contribute to motor responses, the behavioral analysis significantly advances prior study of the navigation algorithm directing fish phototaxis. In particular, the authors demonstrate that reorientation results from the modulatory effects of the detection of relative changes in stimulus intensity on both the turn amplitude and the probability of turning. In addition, they show that turns are directed by a competition between two mechanisms: (i) a stereo-visual bias and (ii) motor persistence. To determine whether a minimal stochastic model is sufficient to account for the overall navigation of real fish, the authors turn to numerical simulations. The results of their simulations reproduce the sensory modulation of transitions between behavioral states as well as orientational biases observed in real fish. Going beyond the mere reproduction of behavioral trends, the authors take advantage of modeling to explore a simple neural-circuit mechanism underlying action selection in the hindbrain.

Essential revisions:

1) The first two paragraphs should be re-written in a manner to better reflect the limits of the eventual conclusions of the paper. Specifically, a speculative hypothesis ("behavior is thus based as a set of statistic (sic) rules that defines how elementary motor motor actions are chained") is stated as fact. That this type of a model well-describes their data is sensible, but to cast all behavior – including non-foraging behavior – in this light seems well outside the scope of what they show here. If they want to make this (in our view, controversial) claim that behavior outside the foraging context all behave in this manner, it should be put forward in the Discussion section as a hypothesis emanating from the work rather than an underlying assumption. Notably, the text in the Discussion section was much more careful in this regard.

2) In Figure 5—figure supplement 1, the authors show three illustrative trajectories produced by the model for different inter-bout intervals. It would be useful and important to provide the reader with a comparison of a set of trajectories of real fish and that simulated by the (full) stochastic model. This would permit the reader to judge the goodness of the reproduction of physical trajectories in addition to the statistical distributions (and density profiles of Figure 4B). At first, one might think that the trajectories shown in Figure 5—figure supplement 1 look more like those of Brownian particles than fish.

3) In their Discussion, the authors argue about the advantages of testing the influence of contrast-driven orientation under constant overall illumination intensities. In the experiments of Figure 3, animals do not experience changes in light intensity as they move with respect to the virtual source. This (simplified) experimental paradigm is assumed to be sufficient to reveal the sensorimotor mechanism directing the klinotaxis/klinokinesis component of the orientation algorithm. Is the stochastic model sufficient to produce the ascent of light gradients in numerical simulations – a behavior more closely related to real-life situations?

---

## [Author Response]

Essential revisions:1) The first two paragraphs should be re-written in a manner to better reflect the limits of the eventual conclusions of the paper. Specifically, a speculative hypothesis ("behavior is thus based as a set of statistic (sic) rules that defines how elementary motor motor actions are chained") is stated as fact. That this type of a model well-describes their data is sensible, but to cast all behavior – including non-foraging behavior – in this light seems well outside the scope of what they show here. If they want to make this (in our view, controversial) claim that behavior outside the foraging context all behave in this manner, it should be put forward in the Discussion section as a hypothesis emanating from the work rather than an underlying assumption. Notably, the text in the Discussion section was much more careful in this regard.

We agree that this claim was over-reaching. We modified the Introduction to immediately narrow the scope of this hypothesis to the locomotion of small animals.

2) In Figure 5—figure supplement 1, the authors show three illustrative trajectories produced by the model for different inter-bout intervals. It would be useful and important to provide the reader with a comparison of a set of trajectories of real fish and that simulated by the (full) stochastic model. This would permit the reader to judge the goodness of the reproduction of physical trajectories in addition to the statistical distributions (and density profiles of Figure 4B). At first, one might think that the trajectories shown in Figure 5—figure supplement 1 look more like those of Brownian particles than fish.

This perception is due to the scale of this particular figure, which encompasses very long sequences (much longer than experimentally measured trajectories). We added a supplementary figure (Figure 5—figure supplement 1) showing both real and simulated trajectories at a larger magnification for which individual bouts are visible. This figure establishes that the circuit-based simulation does correctly capture the fine-scale geometry of the trajectories.

3) In their Discussion, the authors argue about the advantages of testing the influence of contrast-driven orientation under constant overall illumination intensities. In the experiments of Figure 3, animals do not experience changes in light intensity as they move with respect to the virtual source. This (simplified) experimental paradigm is assumed to be sufficient to reveal the sensorimotor mechanism directing the klinotaxis/klinokinesis component of the orientation algorithm. Is the stochastic model sufficient to produce the ascent of light gradients in numerical simulations – a behavior more closely related to real-life situations?

Our aim was to characterize Zebrafish light-seeking strategies in the presence of a distant light source. In this case, the illumination angular profile experienced by the fish is independent of its (x,y) position in space – no spatial gradient – and thus also time-invariant. We showed that the fish is able to orient towards the light source (angular phototaxis), using either the contrast or the total brightness, and is thus able to progress towards it at constant (mean) speed. In a more realistic context, both the contrast and the total brightness would vary with the fish’s body orientation. However, because the two processes were shown to be independent from each other (acting on separate motor variables), these two mechanisms are expected to act in concert leading to efficient positive phototaxis.

When refering to real-life situations, the referee may imply contexts in which the source is located at a finite distance, such that the brightness also varies with the (x,y) spatial position of the fish. The increase in diffusivity induced by light decrement should allow the fish to progress in an illumination spatial gradient (even without contrast), by analogy with bacteria chemotaxis. However, we did not examine this process experimentally, and thus cannot provide quantitative comparison with experimental data. This particular condition is beyond the scope of our work.